# Control of Bromate Formation in Desalinated Seawater Production and Transmission with Ammoniation

**Ali A. Alhamzah [1,2], Abdulrahman S. Alofi [3], Abdulrahman A. Abid [3] and Christopher M. Fellows [1,2,*]**

[1]   Water Technologies Innovation Institute & Research Advancement (WTIIRA), Saline Water Conversion Corporation, Jubail 31951, Saudi Arabia; aal-hamzah2@swcc.gov.sa
[2]   School of Science and Technology, The University of New England, Armidale, NSW 2351, Australia
[3]   Saline Water Conversion Corporation, Riyadh 11547, Saudi Arabia; aalofi@swcc.gov.sa (A.S.A.); aabid@swcc.gov.sa (A.A.A.)
[*]   Correspondence: cfellows@une.edu.au or cmichael@swcc.gov.sa; Tel.: +966-59-417-1150

**Abstract:** Bromate is a potentially carcinogenic disinfection by-product of potential concern in desalinated waters, where bromide derived from seawater can be converted to bromate by the oxidising species used for disinfection. Historically, it has been difficult to maintain complete adherence to national standards of no more than 10 ppb for bromate at all locations served with desalinated seawater by the Saline Water Conversion Corporation (SWCC) in the Kingdom of Saudi Arabia. In this full-scale study, the addition of 100–200 ppb of ammonia to the produced water of a Multi-Stage Flash Desalination plant effectively controlled the formation of bromate in the transmission system supplying inland centres in the Makkah Province of the Kingdom of Saudi Arabia (Arafa, Taif) on a time scale sufficient for the distribution of water to the consumer, even when the bromide content of the produced water was artificially enhanced (up to 132 ppb) via the addition of seawater.

**Keywords:** ammonia; bromate; desalination; multi-stage flash; transmission

## 1. Introduction

A significant public health concern in any system where potable water is stored or transported is the possibility of the formation of disinfection by-products. Disinfection is a necessity to avoid the bacterial contamination of drinking water, but the oxidising species used can generate a range of potentially toxic and carcinogenic species from trace components of the produced water [1–4]. When water contains bromide ions, these disinfection by-products can include the bromate ion as well as brominated organic species such as bromoacetic acid, bromoform, and bromodichloromethane [5,6]. Historically, bromate control has been an issue primarily in surface and ground water treatment with ozonation, and there is a considerable body of research literature addressing control measures for this problem [7–15].

Recent interest in bromate control in the Kingdom of Saudi Arabia in waters treated by chlorination rather than ozonation arises from a unique combination of two factors: (1) increasing use of seawater desalination by reverse osmosis (SWRO), rather than thermal methods, which leads to product water with a higher concentration of bromide; (2) transmission of the majority of produced water over lengthy pipelines (>100 km) at relatively high temperatures, as the formation of disinfection by-products increases with both temperature and time. While SWRO is a well-established technology in many parts of the world, outside of the Kingdom of Saudi Arabia it is used almost entirely to serve coastal centres, where it is consumed in close proximity to its point of production. These two factors have led to challenges in consistently meeting the 10 ppb maximum limit set by regulatory authorities in the Kingdom of Saudi Arabia [16,17].

This work reports on efforts by the Saline Water Conversion Corporation (SWCC) to minimize the formation of bromate and related disinfection by-products in the water transmitted to consumers via the addition of ammonia to produced desalinated water. We will first review and discuss the mechanism of bromate formation in water treated by chlorination and the probable role of ammonia in affecting this mechanism will in the light of SWCC's experience in monitoring water quality. The details of the application of ammonia at a SWCC desalination plant at Shoiabah on the Red Sea and the results of this application will then be presented.

## 2. Mechanisms and Kinetics

The mechanism of the formation of bromate under chlorination is well understood, with the first step being the formation of hypochlorite.

$$Cl_2 + H_2O \rightarrow HClO + H^+ + Cl^- \tag{1}$$

$$Cl_2 + H_2O \rightarrow ClO^- + 2H^+ + Cl^- \tag{2}$$

At the temperatures of interest in the storage and transmission of product water in the Kingdom of Saudi Arabia (15–45 °C), reaction (2) is dominant, and most of the dissolved oxidant is hypochlorite, rather than hypochlorous acid [18,19].

The second step is the oxidation of bromide to hypobromite under basic (reaction (3)) or acidic (reaction (4)) conditions.

$$ClO^- + Br^- \rightarrow BrO^- + Cl^- \tag{3}$$

$$HClO + Br^- \rightarrow HBrO + Cl^- \tag{4}$$

Both reactions are thermodynamically favourable over the temperature range of interest. However, the reaction rate for the reaction under acidic conditions is about $10^6$ times higher, and the reaction is acid-catalyzed [20]. It is intuitive that this should be the more favourable reaction because it does not involve bringing two anions together. Thus, the reaction will be dominated by reaction (4) even at relatively high pH and the reaction under basic conditions can be ignored over the pH range of interest.

The third step is the oxidation of hypobromite to bromate. There is a significant body of literature on mechanisms for this reaction under ozonation conditions, but relatively little work has been conducted on the mechanism in the presence of an oxidant. Stoichiometrically, this is reported to occur by the disproportionation of hypobromous acid to give bromate and bromide [21,22]:

$$3HBrO \rightarrow HBrO_3 + 2HBr \tag{5}$$

Trimolecular reactions are statistically implausible, so this process is likely to proceed in two steps, for example, as proposed by Margerum and Huff Hartz [23]:

$$2HOBr \rightarrow BrO_2^- + Br^- + 2H^+ \ (pK_a\ HBrO_2 = 6.25) \tag{6}$$

Followed by:

$$HOBr + BrO_2^- \rightarrow BrO_3^- + Br^- + H^+ \ (pK_a\ HBrO_3 = -2) \tag{7}$$

A similar second step for the formation of bromate from hypobromite has been previously reported with ozone as an oxidant [24]. Margerum and Huff Hartz found that reaction (5) was second order in HOBr in the presence of HOCl, and thus proposed [23]:

$$2HOBr \rightarrow BrO_2^- + Br^- + 2H^+ \ k = 0.015\ M^{-1}s^{-1} \tag{8}$$

$$HOCl + BrO_2^- \rightarrow BrO_3^- + Cl^- + H^+ \ k = \text{fast} \tag{9}$$

with the first step clearly being rate limiting. The apparent rate coefficient $k$ increased with total chlorine, but this only became evident above 5 mM (>175 ppm) Cl; Margerum and Huff Hartz reported that the rate constant for the second-order decomposition of HOBr has a maximum at pH ~7.2 and fit this with a complex mechanistic model, but this maximum is dependent on only one data point at pH 7.6 [23]. In contrast to this, we have consistently observed more bromate formation at higher pH over the range of pH values seen in product water (7.5–9.0) in routine assessments of water quality in the SWCC production and transmission systems. Therefore, we postulate a significant contribution of the cross-reaction between hypobromite and hypobromous acid:

$$HOBr + BrO^- \rightarrow BrO_2^- + Br^- + H^+ \tag{10}$$

This reaction should be favourable since it requires less charge separation overall, and if it goes at a slower rate than the 2HOBr reaction, it could contribute negligibly at pH ~7 but give elevated rates any higher pHs, as observed in the SWCC transmission system, for example, if the HOBr + BrO$^-$ reaction proceeds at one-third the rate of the 2HOBr reaction (Figure 1).

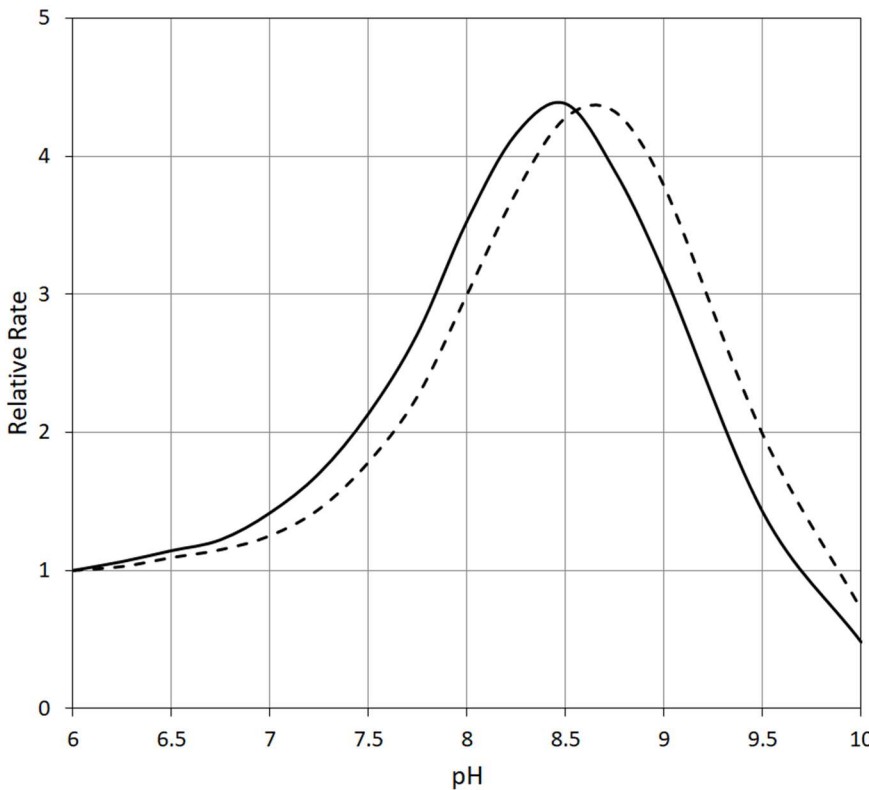

**Figure 1.** Modelled relative rate of the generation of hypobromate with pH at T = 20 °C (dashed line) and 40 °C (solid line), assuming the rate limiting step is the bimolecular reaction of HOBr with HOBr (relative $k$ = 3) or HOBr with OBr$^-$ (relative $k$ = 1).

The overall scheme of bromate formation with estimated kinetic and thermodynamic parameters of importance is summarized below (Scheme 1).

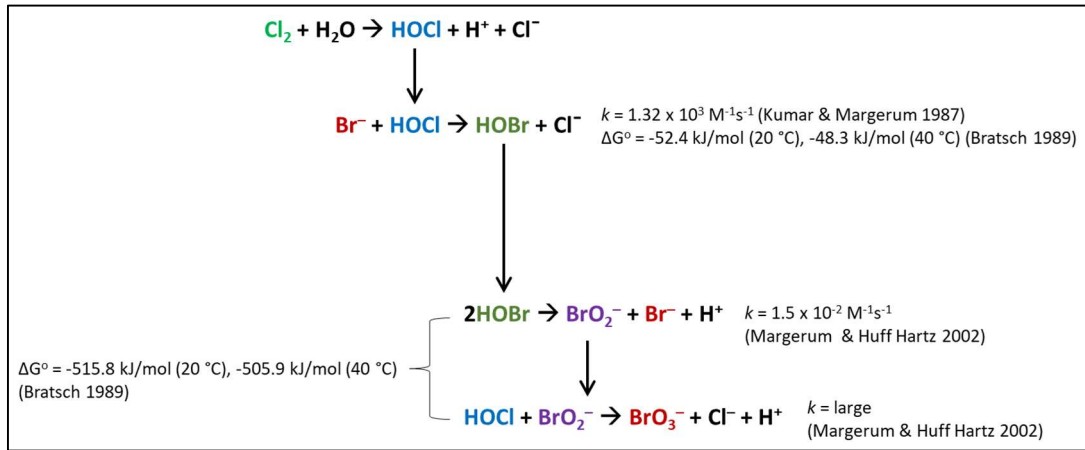

**Scheme 1.** Pathways of formation of bromate from bromide with chlorine oxidant [20,23,25].

Ammonia can initially impact the system as outlined above (Scheme 1) in two primary ways [24]:

(i)　Reducing the amount of HOCl available for the oxidation of bromide [26]:

$$NH_3 + HOCl \rightarrow NH_2Cl + H_2O \ (k = 4.2 \times 10^6 \ M^{-1}s^{-1}) \tag{11}$$

(ii)　Reacting with HOBr so it is unavailable for oxidation to bromate [27]:

$$NH_3 + HOBr \rightarrow NH_2Br + H_2O \ (k = 7.5 \times 10^7 \ M^{-1}s^{-1}) \tag{12}$$

Given the relative concentrations expected of HOCl and HOBr, reaction (11) is likely to be more important than reaction (12) despite the order of magnitude difference in rates. Because of the impact of ammonia on reducing the amount of HOCl (while not losing disinfectant capacity, as monochloroamine is also a disinfectant), Ling et al. recommended the addition of ammonia before chlorination in ozonation reactions [24].

These main reactions removing HOBr and hence reducing bromate formation are shown below (Scheme 2).

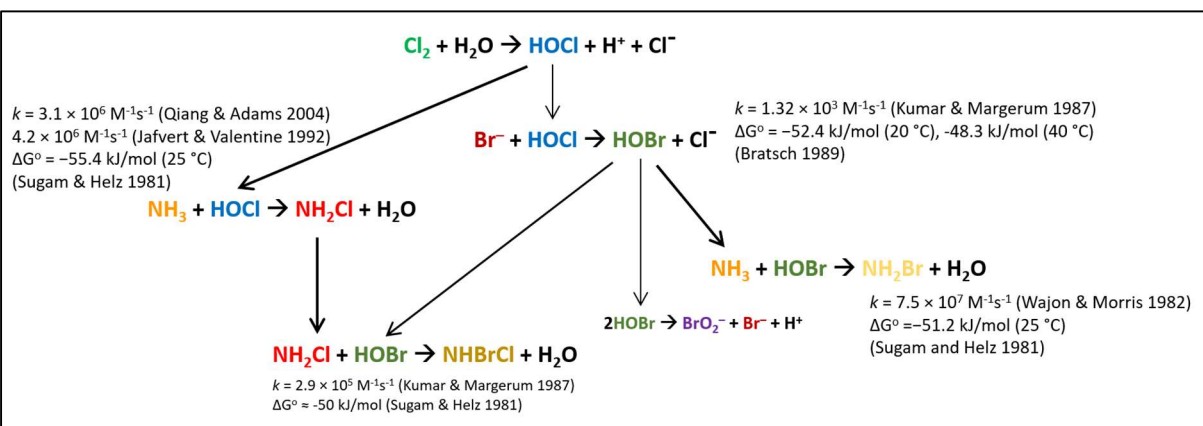

**Scheme 2.** Pathways for inhibition of bromate from bromide with chlorine oxidant [20,25,27–29].

The monochloramine and monobromamine can react further [30]:

$$NH_2Cl + HOBr \rightarrow NHBrCl + H_2O \ (k = 2.9 \times 10^5 \ M^{-1}s^{-1}). \tag{13}$$

Meanwhile, $NH_2Br$ may disproportionate reversibly in a base-catalyzed reaction [31]:

$$2NH_2Br \leftrightharpoons NHBr_2 + NH_3 \text{ (}K\text{ of order 0.5-5 depending on catalyst)} \tag{14}$$

Following that, it will then react with $NH_2Cl$ [32]:

$$NHBr_2 + NH_2Cl \rightarrow NH_2Br + NHBrCl \tag{15}$$

These haloamines may also react with any remaining bromide present in the system, giving bromamine [33],

$$NH_2Cl + Br^- \rightarrow NH_2Br + Cl^- \text{ (}k = 0.014 \text{ M}^{-1}\text{s}^{-1}\text{)} \tag{16}$$

bromochloramine [33,34],

$$2NH_2Cl + Br^- \rightarrow NHBrCl + NH_3 + Cl^- \text{ (}k = 2.9\text{–}3.4 \times 10^6 \text{ M}^{-2}\text{s}^{-1}\text{)} \tag{17}$$

or dibromamine [24,35]:

$$NHBrCl + Br^- \rightarrow NHBr_2 + Cl^- \text{ (}k = 565 \text{ M}^{-1} \text{ s}^{-1}\text{)} \tag{18}$$

The net result of these reactions should be the sequestration of the initially present bromide as haloamines, predominantly as NHBrCl [33,36]. The net of reactions of the haloamines is summarized below (Scheme 3).

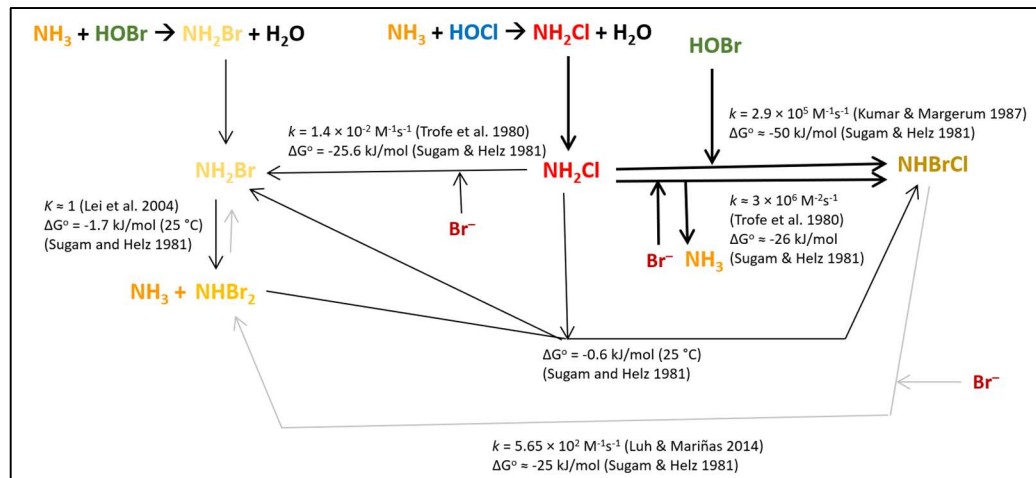

**Scheme 3.** Interaction of haloamine reactions in ammoniated and chlorinated waters containing bromide [20,29,31,33,35].

NHBrCl has been reported to be less reactive than the mono- and dibromamines [37], giving an expected impact on the yield of halogenated organic contaminants, as well as bromate (pace Valentine [38]). It is not wise to trust an experimental result until it has been validated by theory [39], and the experimental observation is supported by the free energies calculated for bromamines and chloramines in aqueous solution by Trogolo and Aret [40]. These free energy values suggest that for a generic bromination reaction,

$$A + NH_xBr_y(Cl) \rightarrow ABr + NH_{x-1}Br_{y-1}(Cl) \tag{19}$$

the free energy of the reaction will be higher for $NHBrCl$ than for $NH_2Br$ (2.5 kJ/mol less) and $NHBr_2$ (5.5 kJ/mol less). Thus, the ammoniation before chlorination process can be expected to strongly reduce the incidence of other brominated disinfection products. The addition of ammonia before chlorination will cause the formation of increased levels of chloramine. Ammoniation after chlorination is more problematic, because if the rate

coefficients quoted above are correct, this will lead to a significant population of $NH_2Br$ and $NHBr_2$, which are good brominating agents [41,42]. However, minimizing the time between chlorination and ammoniation to approximately 30 s has also been shown to give primarily NHBrCl [43]. Sun et al. reported in a similar system to desalination (secondary effluent chlorination) that trihalomethane (THM) production was reduced but haloacetic acid concentrations were increased at a bromide concentration of 140 ppb and not reduced at 50 ppb; it is unclear from the description of this study whether ammonia was added before or after chlorination [44]. This is consistent with other reports of reduced THM yields under conditions where bromamines replace hypobromous acid [29] and that bromamines are more reactive in forming haloacetic acids [41].

As long as there is residual chlorine in the system, it is expected that this chlorine will continue to generate dibromochloramine. Only when there is no remaining oxidant is $NBr_2Cl$ expected to decompose in aqueous solution to $H^+$, $N_2$, $Cl^-$, $Br^-$, and $BrO^-$ [35]. Luh and Mariñas (2014) studied the decomposition of $NH_2Cl$ with a large excess (by 5 to 50 times) of $Br^-$ over $NH_2Cl$ and clearly observed the replacement of $NH_2Cl$ by NHBrCl, which decomposed at a slower rate [35]. At the lower relative concentrations of bromide in desalination product water (~100 ppb), this suggests that the bromide-consuming reactions discussed here will not deplete the $NH_2Cl$ available. Hu et al. calculated for a system also containing relatively high concentrations of bromide, but with an excess of chloramine over bromide (0.05 mM NH2Cl, 1.6–3.2 ppm 0.02/0.04 mM $Br^-$), that almost all bromine would be present as NHBrCl, and found it to be rapidly degraded by CuO [45].

The body of data obtained up until now and outlined here on the mechanism and kinetics of the interactions between ammonia, chlorine, and bromide therefore suggests that chlorination followed by ammoniation of the water produced by seawater desalination is a potential strategy for controlling the formation of bromate and other disinfection by-products of concern.

## 3. Materials and Methods

Water samples were collected from the Shoaibah Phase 2 desalination plant of the SWCC and from a number of storage tanks adjacent to the plant, pumping stations on transmission lines originating at the Shoaibah desalination complex, and storage tanks in communities near the termini of these transmission lines (Arafa and Taif), in 2022 and 2023. The production and transmission system is complex, and a simplified diagram showing the relationship between the different sampling points is given below (Figure 2).

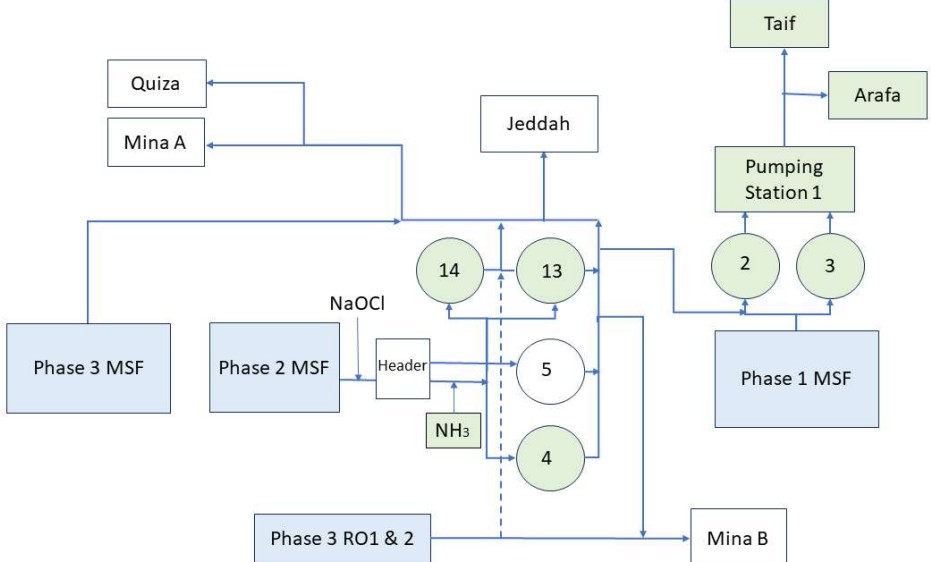

**Figure 2.** Sampling points in the Shoaibah desalination plant campus and associated transmission system.

Analysis of the parameters of interest was carried out within three weeks of collection in the laboratories of the Desalination Technologies Research Institute of the SWCC in Al Jubail, Kingdom of Saudi Arabia, using standard methods as follows:

pH: Potentiometry using a standard hydrogen electrode [46];

Ammonium, bromide, bromate, chloride, and nitrate: Ion chromatography with chemical suppression of eluent conductivity [47].

Trihalomethanes and other organic contaminants: EPA 524 (Gas chromatography-Mass spectrometry, GC-MS). One sample (5 May 2022) was analyzed in the WTIIRA laboratories; one sample (25 May 2023) by ALS Arabia, Dammam, Saudi Arabia; and one sample (6 June 2023) by SGS Inspection Services, Jubail, Saudi Arabia.

Ammonia was added to an existing post-treatment system utilising chlorination via sodium hypochlorite. The ammonia addition point was located about 50 m downstream of the chlorination point.

Two rounds of trials were made, in 2022 and 2023.

The 2022 trials were carried out in several stages:

(0)   Addition of sufficient seawater to bring the TDS up to 82–94 ppm, which should have given bromide concentrations in the range 49–72 ppb if there was no selective rejection or permeation of bromide (14–16 April 2022);

(1)   Addition of 120 ppb $NH_3$ along with sufficient seawater to bring the TDS up to 81–89 ppm, which should have given bromide concentrations in the range 47–64 ppb (17–19 April 2022);

(2)   Addition of 120 ppb $NH_3$ along with sufficient seawater to bring the TDS up to 106–115 ppm, (96–113 ppb $Br^-$) (20–25 April 2022);

(3)   Addition of 200 ppb $NH_3$, along with sufficient seawater to bring the TDS up to 110–132 ppm (104–132 ppb $Br^-$) (26 April–3 May 2022);

(4)   Addition of 200 ppb $NH_3$ and sufficient sodium hydroxide to adjust the pH to 8.7, along with sufficient seawater to give a TDS of 114 ppm (112 ppb bromide) (4–5 May 2022).

For a number of locations, control data were also collected on 12–13 April 2022.

A second series of tests was carried out in 2023, with the application of chlorination/ammoniation in both Phase 1 and Phase 2 of the Shoaibah Desalination plant:

(1)   Addition of 100 ppb $NH_3$ only (23 April–5 May 2023);

(2)   Addition of 100 ppb $NH_3$ along with sufficient seawater to bring the TDS up to 119–242 ppm, which should have given bromide concentrations in the range 116–325 ppb. (4–14 May 2023);

(3)   Addition of 100 ppb $NH_3$, along with sufficient seawater to bring the TDS up to 130–230 ppm, (134–305 ppb $Br^-$) (15 May–17 June 2023);

(4)   Addition of 100 ppb $NH_3$ along with sufficient seawater to bring the TDS up to 82–125 ppm (52–126 ppb $Br^-$) (19 June–12 July 2023);

(5)   Addition of 90 ppb $NH_3$ along with sufficient seawater to bring the TDS up to 82–129 ppm (52–133 ppb $Br^-$) (13 July–7 Aug 2023).

As each phase in the 2022 and 2023 test series corresponds to multiple measurements of a number of days, the average value and an error value calculated from the standard deviation in the measured values are reported for each measured parameter for each phase.

## 4. Results and Discussion

The key parameters of interest in both the 2022 and 2023 trials were the concentrations of bromide, bromate, ammonium, and nitrate in the product waters. Bromide and bromate were taken as the critical input and output of the complex system outlined in the introduction, and ammonium and nitrate were taken as potentially problematic by-products of the ammonia added to the system.

*4.1. 2022 Trials*

The measured bromide concentrations before chlorination and ammoniation were significantly below the predicted values, with $87 \pm 4$ ppb measured during stages 3 and 4. These were also below the value that could be predicted from measurements of the chloride concentration in the water ($99 \pm 3$ ppb) using the expected ratio of Cl:Br in seawater. As the absolute number of bromine-containing species is less important than their relative proportions, the correlation between chloride and bromide concentration was used to estimate the total bromine concentration in product water samples after chlorination and compared to the amount of bromide.

After chlorination and ammoniation, the treated water was passed on to tanks within the Shoaibah desalination plant complex, which also receive water from other sources, some of which produce low-TDS water with no significant bromate problems, while others are more problematic; these in turn proceed to pumping stations which draw upon multiple tanks.

While the net of reactions described in Scheme 3 suggests that very little free bromide will be observed, being sequestered as bromamines if it is not present as hypobromite ion/hypobromous acid, ion chromatography found significant amounts of bromide ions in all waters investigated. In waters treated with ammonia, concentrations of bromide were consistently higher. Unless there are serious issues with the mechanisms postulated in the literature, this suggests that the bromamines are not stable under the conditions of ion chromatography. High levels of bromide in IC have also been observed by Pearce et al. in studies where high concentrations of chloramine were added to product water [48].

From Table 1, it can be seen that the apparent ratio of non-bromide bromine-containing species (presumably primarily $BrO^-$) as a fraction of the total bromine in the stream decreases on addition of 120 ppm ammonia (cf. the values obtained in Tank 5 with Tanks 4, 13, and 14), decreasing further as ammonia concentrations are increased to 200 ppm, and decreasing further again when pH is increased.

**Table 1.** Estimated % of non-bromide bromine at locations within the Shoaibah desalination plant site and associated transmission system, 2022. Values **bold and underlined** are locations where water quality should be affected by ammoniation.

| Stage | Control | 0 | 1 | 2 | 3 | 4 |
|---|---|---|---|---|---|---|
| Line B after chlorination, before ammoniation | | $72 \pm 4$ | | | | |
| Header 1 | | | | | $58 \pm 22$ | $8 \pm 4$ |
| Header 2 | | | | | **$18 \pm 22$** | **$6 \pm 6$** |
| Tank 4 | | $70 \pm 2$ | **$53 \pm 10$** | **$21 \pm 23$** | **$8 \pm 4$** | **$0$** |
| Tank 5 | | $53 \pm 12$ | $61 \pm 9$ | $54 \pm 20$ | $43 \pm 21$ | $63 \pm 3$ |
| Tank 13 | $73 \pm 5$ | $74 \pm 2$ | **$37 \pm 28$** | **$48 \pm 22$** | **$5 \pm 2$** | **$2 \pm 4$** |
| Tank 14 | | $67 \pm 6$ | **$44 \pm 29$** | **$46 \pm 26$** | **$5 \pm 5$** | **$0$** |
| PS (Pumping Station) 1A | $0$ | $4$ | $7 \pm 4$ | $6 \pm 1$ | $3$ | $0$ |
| PS1B | $61 \pm 4$ | $43$ | **$68 \pm 7$** | **$55 \pm 12$** | **$8 \pm 10$** | **$0$** |
| Jeddah PS | $63 \pm 1$ | $44$ | $61 \pm 17$ | $59 \pm 17$ | $56 \pm 13$ | $49$ |
| Quiza PS | $47 \pm 14$ | $58$ | $65 \pm 13$ | $70 \pm 10$ | $66 \pm 11$ | $23$ |
| Mina A PS | $47 \pm 27$ | $53$ | $49 \pm 36$ | $66 \pm 12$ | $35 \pm 8$ | $60$ |
| Mina B PS | $46 \pm 26$ | $72$ | $54 \pm 12$ | $61 \pm 15$ | $55 \pm 17$ | $78$ |

From other indicators (F, $SO_4$, Ca) the water sampled in PS-1A was not derived from the water treated by chlorination–ammoniation during this time, but from the Phase 1 MSF plant (Figure 2). No residual $NH_4$ was measured in this water at any time during the study.

Note the clear fall in non-bromide bromine in Tanks 4, 13, and 14 and pumping station 1B over the time of the test: a small reduction at 100 ppb $NH_3$ treatment, followed by a larger fall at 200 ppb $NH_3$ and a larger fall again when pH was adjusted upwards.

Bromate reduction is not clearly significant with 120 ppb NH₃ addition (stages 1 and 2) but is marked at 200 ppb bromate addition (stages 3 and 4) (Table 2).

**Table 2.** Measured bromate ion concentration (ppb) at locations within the Shoiabah desalination plant site and associated transmission system, 2022. Values **<u>bold and underlined</u>** are locations where water quality should be affected by ammoniation.

| Stage | Control | 0 | 1 | 2 | 3 | 4 |
|---|---|---|---|---|---|---|
| Line B after chlorination, before ammoniation | | 0 | | | | |
| Header 1 | | | | | 0 | 0 |
| Header 2 | | | | | **<u>0</u>** | **<u>0</u>** |
| Tank 4 | | 7 ± 1 | **<u>5 ± 2</u>** | **<u>4</u>** | **<u>1 ± 1</u>** | **<u>0</u>** |
| Tank 5 | | 12 ± 4 | 7 ± 2 | 9 ± 2 | 14 ± 2 | 16 ± 1 |
| Tank 13 | 2 ± 1 | 2 ± 1 | **<u>2 ± 1</u>** | **<u>2 ± 1</u>** | **<u>0</u>** | **<u>0</u>** |
| Tank 14 | | 2 ± 1 | **<u>2 ± 1</u>** | **<u>1 ± 1</u>** | **<u>0</u>** | **<u>0</u>** |
| PS (Pumping Station) 1A | 0 | 2 | 2 | 1 ± 1 | 0 | 0 |
| PS1B | 2 | 2 | **<u>2</u>** | **<u>1 ± 1</u>** | **<u>0</u>** | **<u>0</u>** |
| Jeddah PS | 3 ± 1 | 3 | 2 | 2 | 1 | 3 ± 1 |
| Quiza PS | 6 ± 1 | 4 | 6 ± 1 | 5 ± 2 | 5 | 4 |
| Mina A PS | 6 ± 1 | 4 | 5 ± 1 | 4 ± 1 | 6 | 6 ± 1 |
| Mina B PS | 5 ± 1 | 4 | 4 ± 1 | 2 ± 1 | 4 ± 1 | 5 ± 1 |

The day-by-day bromate data for the tanks containing untreated water (Tank 5) and primarily treated water (Tanks 4, 13, and 14) are displayed in Figure 3.

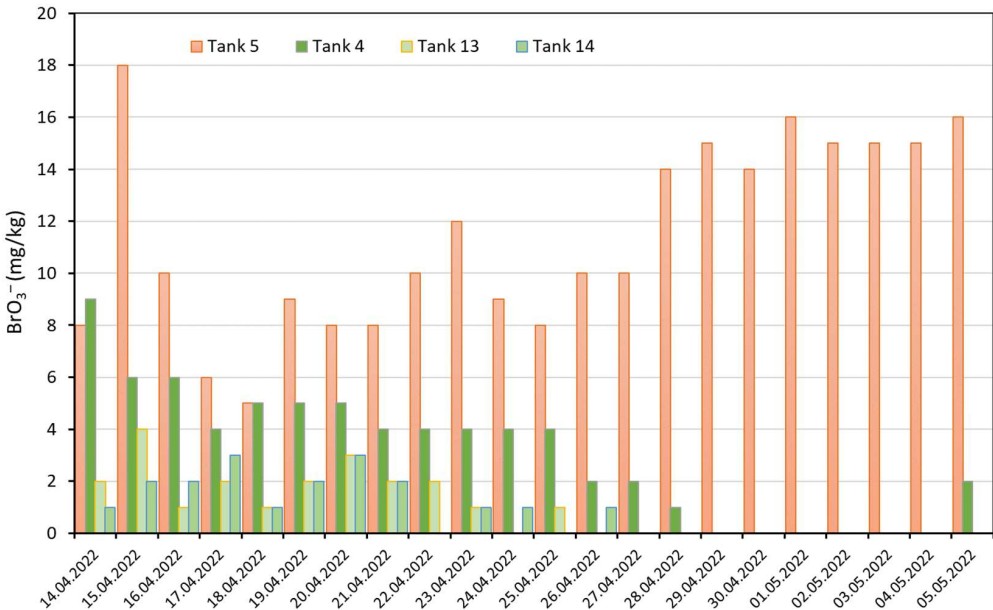

**Figure 3.** Bromate concentration in Shoiabah storage tanks during the 2022 trials.

Ammonium concentration should be a good indicator for the presence of treated water. It can be seen that there is essentially no ammonium at any of the pumping stations and disproportionately greater quantities of residual ammonium with 200 ppb treatment (Table 3). See, for example, Tank 4 at stage 2, where the effect of the treatment appears to be significant in terms of the bromide/hypobromite ratio and the amount of bromate observed. It is also clear that the analysis is not quantitative, as concentrations of ammonium up to twice the predicted concentration of the ammonia added are observed in some instances. There are also occasional outliers with no clear explanation at locations where ammonium

should not be present, and an absence of ammonium under conditions (Tank 4, stages 1–2) where it would be expected; there are thus concerns about the methodology of ammonium determination.

**Table 3.** Measured ammonium ion concentration (ppb) at locations within the Shoiabah desalination plant site and associated transmission system, 2022. Values **<u>bold and underlined</u>** are locations where water quality should be affected by ammoniation.

| Stage | Control | 0 | 1 | 2 | 3 | 4 |
|---|---|---|---|---|---|---|
| Line B after chlorination, before ammoniation | | 0 | | | | |
| Header 1 | | | | | 14 ± 23 | 0 |
| Header 2 | | | | | **282 ± 118** | **403 ± 107** |
| Tank 4 | | 0 | **<u>0</u>** | **<u>0</u>** | **<u>162 ± 73</u>** | **<u>306 ± 10</u>** |
| Tank 5 | | 0 | 0 | 0 | 12 ± 31 | 0 |
| Tank 13 | 0 | 0 | **<u>36 ± 29</u>** | **<u>6 ± 14</u>** | **<u>123 ± 58</u>** | **<u>180 ± 31</u>** |
| Tank 14 | | 0 | **<u>40 ± 32</u>** | **<u>24 ± 26</u>** | **<u>140 ± 60</u>** | **<u>200 ± 28</u>** |
| PS (Pumping Station) 1A | 0 | 0 | 0 | 0 | 0 | 62 |
| PS1B | 0 | 0 | **<u>0</u>** | **<u>16 ± 23</u>** | **<u>0</u>** | **<u>0</u>** |
| Jeddah PS | 20 ± 21 | 0 | 0 | 0 | 0 | 0 |
| Quiza PS | 0 | 0 | 0 | 0 | 0 | 0 |
| Mina A PS | 0 | 0 | 20 ± 30 | 0 | 0 | 0 |
| Mina B PS | 0 | 0 | 0 | 0 | 0 | 0 |

One concern about addition of ammonia is its potential oxidation to nitrate, with implications for the quality of water delivered to the consumer. The ppb of nitrate present in the system was also assessed (Table 4). There is no clear relationship between the amount of nitrate observed and the amount of ammonia added to the system. Although nitrate levels in Tank 5, not receiving the treated water, were assessed as lower than Tanks 4, 13, and 14 in phases 2 and 3, comparable values were obtained for all four tanks in phase 4.

**Table 4.** Measured nitrate ion concentration (ppb) at locations within the Shoiabah desalination plant site and associated transmission system, 2022. Values **<u>bold and underlined</u>** are locations where water quality should be affected by ammoniation.

| Stage | Control | 0 | 1 | 2 | 3 | 4 |
|---|---|---|---|---|---|---|
| Line B after chlorination, before ammoniation | | 29 ± 4 | | | | |
| Header 1 | | | | | 32 ± 5 | 28 |
| Header 2 | | | | | **<u>32 ± 4</u>** | **<u>25</u>** |
| Tank 4 | | 27 ± 1 | **<u>34 ± 4</u>** | **<u>40 ± 7</u>** | **<u>44 ± 5</u>** | **<u>39 ± 1</u>** |
| Tank 5 | | 16 ± 2 | 31 ± 14 | 24 ± 6 | 26 ± 6 | 31 |
| Tank 13 | 35 ± 1 | 20 ± 4 | **<u>32 ± 10</u>** | **<u>39 ± 5</u>** | **<u>38 ± 3</u>** | **<u>28 ± 2</u>** |
| Tank 14 | | 25 ± 3 | **<u>35 ± 6</u>** | **<u>41 ± 11</u>** | **<u>39 ± 4</u>** | **<u>28 ± 1</u>** |
| PS (Pumping Station) 1A | 53 ± 29 | 14 | 38 ± 13 | 35 ± 3 | 42 ± 4 | 32 |
| PS1B | 29 ± 3 | 53 | **<u>27 ± 4</u>** | **<u>32 ± 5</u>** | **<u>37 ± 4</u>** | **<u>53</u>** |
| Jeddah PS | 31 ± 1 | 33 | 25 ± 1 | 34 ± 6 | 36 ± 4 | 30 |
| Quiza PS | 33 | 29 | 28 ± 1 | 29 ± 2 | 34 ± 1 | 33 |
| Mina A PS | 45 ± 14 | 30 | 34 ± 8 | 30 ± 1 | 35 ± 2 | 26 |
| Mina B PS | 9 ± 5 | 16 | 21 ± 5 | 15 ± 5 | 15 ± 3 | 19 |

The data on sites taking water not treated by this protocol illustrate the range in values found within the systems due to variations in the quality of the water produced, the mixing

of these waters in different proportions, and the different environmental factors affecting these waters as they move through the transmission system.

Under the phase 4 trial conditions, samples of water from different parts of the Shoiabah plant were analyzed for trihalomethanes (Table 5). Trihalomethanes were only detected in the waters that were chlorinated and not ammoniated. Note that these species were most likely generated during the time when the samples were transported to the laboratory for analysis.

**Table 5.** Measured trihalomethanes concentration (ppb) at locations within the Shoiabah desalination plant site and associated transmission system on 5 May 2022.

| Stage | $CHCl_3$ | $CHCl_2Br$ | $CHClBr_2$ | $CHBr_3$ |
|---|---|---|---|---|
| Line B before chlorination | <1 | <1 | <1 | <1 |
| Line B after chlorination, before ammoniation | 3 | 9 | 18 | 13 |
| Header 1 | <1 | <1 | <1 | <1 |
| Tank 13 | <1 | <1 | <1 | <1 |
| Tank 14 | <1 | <1 | <1 | <1 |

*4.2. 2023 Trials*

As in 2022, a significant reduction in non-bromide bromine was seen in 2023 in the tanks being fed by the Phase 2 Shoiabah desalination plant where the combined chlorination/ammoniation post-treatment was employed, as well as at in pumping station 1 (Table 6) Reductions in bromate at these tanks were also observed (Table 7). Data are given to compare water further down the transmission line not deriving from the treated water (at the PS Jeddah, Quiza, Mina A and Mina B) and also for water deriving from the treated water. The impact of the treatment on the proportion of non-bromide bromine appears to be clear at tanks (Arafa and Taif) located hundreds of kilometres from the water production site. While there is also a dramatic reduction in non-bromide bromine at Mina B pumping station between the control period, which did not receive the ammoniated water, taking the values in aggregate it is clear that this arises from an outlier in the upwards direction during the control period.

**Table 6.** Estimated % of non-bromide bromine at locations within the Shoiabah desalination plant site and associated transmission system, 2023. Values **bold and underlined** are locations where water quality should be affected by ammoniation.

| Stage | Control | 1 | 2 | 3 | 4 |
|---|---|---|---|---|---|
| Phase 1–Tank 2 | 18 ± 31 | **15 ± 21** | **11 ± 13** | | |
| Phase 1–Tank 3 | 43 ± 11 | **0 ± 11** | **0 ± 7** | 8 ± 8 | |
| Tank 4 | 36 ± 10 | **3 ± 13** | **26 ± 19** | **35 ± 18** | |
| Tank 13 | 50 ± 13 | **11 ± 9** | **7 ± 27** | **23 ± 25** | |
| Tank 14 | 42 ± 14 | **3 ± 15** | **13 ± 19** | **30 ± 13** | |
| PS 1 | 49 | **18 ± 14** | **24 ± 13** | **27 ± 16** | **8 ± 35** |
| Jeddah PS | 25 | 39 ± 9 | 33 ± 23 | 34 ± 26 | |
| Quiza PS | 42 | 31 ± 9 | 28 ± 8 | 40 ± 23 | |
| Mina A PS | 50 | 54 ± 11 | 55 ± 12 | 61 ± 9 | |
| Mina B PS | 96 | 33 ± 6 | 28 ± 7 | 56 ± 5 | |
| Arafa Tank Outlet | 52 ± 17 | **11 ± 7** | **12 ± 10** | **30 ± 26** | **36 ± 22** |
| Taif Tank Outlet | 38 ± 9 | **19 ± 3** | **20 ± 15** | **20 ± 15** | |

**Table 7.** Measured bromate ion concentration (ppb) at locations within the Shoiabah desalination plant site and associated transmission system, 2023. Values **<u>bold and underlined</u>** are locations where water quality should be affected by ammoniation.

| Stage | Control | 1 | 2 | 3 | 4 | 5 |
|---|---|---|---|---|---|---|
| Header–Phase 2 | | | | | **<u>0</u>** | **<u>0.17 ± 0.37</u>** |
| Phase 1–Tank 2 | 1 ± 1 | **<u>1 ± 3</u>** | **<u>0</u>** | | | |
| Phase 1–Tank 3 | 1 ± 1 | **<u>0</u>** | **<u>0</u>** | **<u>0</u>** | **<u>0</u>** | **<u>0</u>** |
| Tank 4 | 3 ± 2 | **<u>0</u>** | **<u>0</u>** | **<u>0</u>** | | |
| Tank 13 | 2 ± 1 | **<u>0</u>** | **<u>0</u>** | **<u>0</u>** | | |
| Tank 14 | 1 ± 1 | **<u>0</u>** | **<u>0</u>** | **<u>0</u>** | **<u>0</u>** | **<u>0</u>** |
| PS 1 | 0 | **<u>0</u>** | **<u>0</u>** | **<u>0</u>** | **<u>0.4 ± 0.7</u>** <br> **<u>0.17 ± 0.37</u>** | **<u>1.4 ± 2.0</u>** |
| Jeddah PS | 0 | 1 ± 1 | 0 | 3 ± 3 | | |
| Quiza PS | 3 | 4 ± 1 | 1 ± 1 | 2 ± 2 | | |
| Mina A PS | 4 | 6 ± 1 | 5 ± 1 | 3 ± 1 | | |
| Mina B PS | 2 | 2 ± 2 | 0 | 2 ± 2 | | |
| Arafa Tank Outlet | 8 ± 1 | **<u>1 ± 1</u>** | **<u>0</u>** | **<u>0</u>** | **<u>2.5 ± 1.8</u>** | |
| Taif Tank Outlet | 8 ± 1 | **<u>2 ± 2</u>** | **<u>0</u>** | **<u>0</u>** | | |

Note that in stages 4 and 5 of the trial, a different set of parameters was measured, and this fact was not realized until the data sets were analyzed later.

The bromate results clearly show minimization of bromate in Tanks 3, 13, and 14, Pumping Station 1, and the Arafa and Taif tanks under all treatment conditions (Figure 4. The bromate values seen under condition 1 at the Arafa and Taif tanks reflect the time delay in transmitting water through the network; if one additional day is included in the control period, the Taif Tank values become 8 ± 2 for the control period and 0 for condition 1. This is the most significant result of this study: the demonstration that the procedure of sequential chlorination and ammoniation can effectively control bromate formation in transmitted water containing up to 300 ppb bromide which has been transported hundreds of kilometres at significant temperatures (average daytime maxima in Jeddah 34/37 °C April/May, in Taif 29/33 °C April/May). A bromate concentration below the target of 10 ppb was observed at Taif under the conditions prevailing in June/July (average daytime maxima Jeddah 38/39 °C, Taif 35/35 °C).

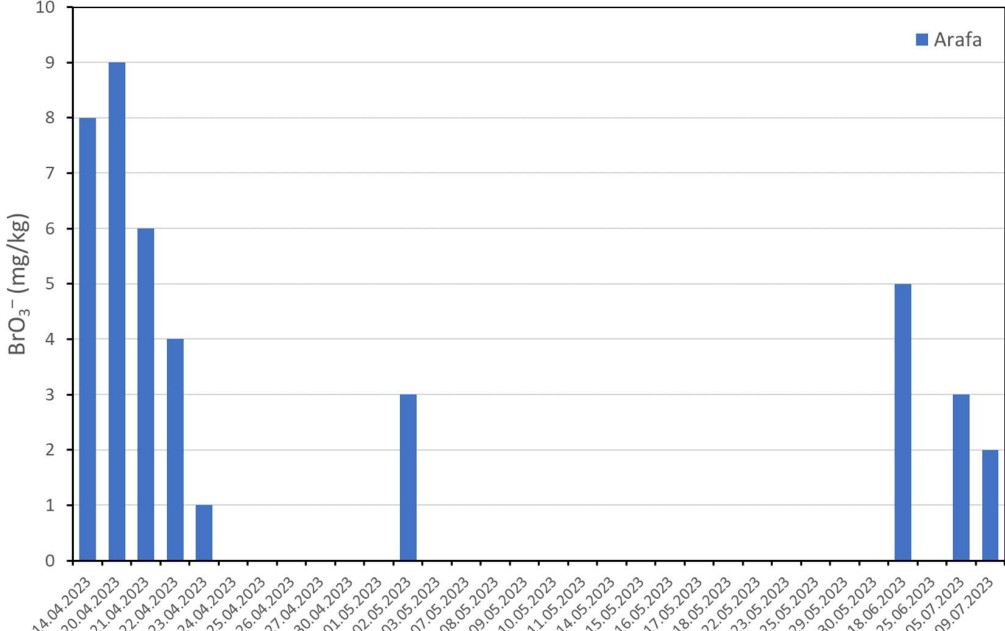

**Figure 4.** *Cont.*

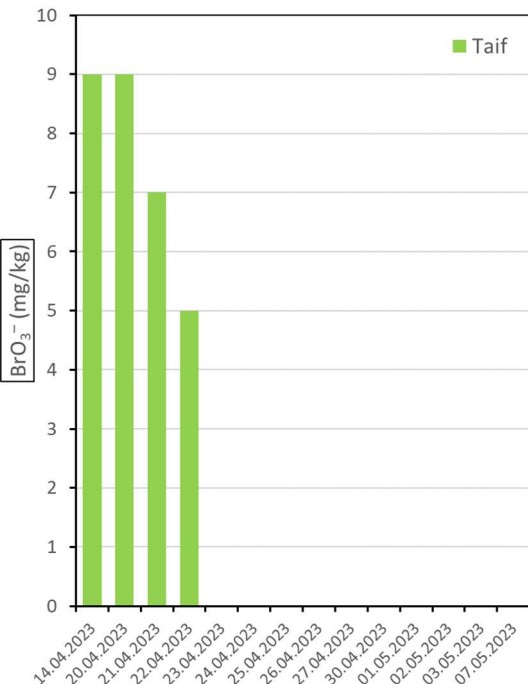

**Figure 4.** Bromate concentration in Arafa and Taif storage tanks during the 2023 trials.

The ammonium results obtained imply that significant amounts of ammonia are present in parts of the system that should not be affected by the trial, while at the same time low values are obtained at the Arafa and Taif tanks where excellent control of bromate was demonstrated (Table 8). Unfortunately, these results thus convey primarily concern about the methodology used to analyze for ammonium.

**Table 8.** Measured ammonium ion concentration (ppb) at locations within the Shoiabah desalination plant site and associated transmission system, 2023. Values **<u>bold and underlined</u>** are locations where water quality should be affected by ammoniation.

| Stage | Control | 1 | 2 | 3 | 4 | 5 |
|---|---|---|---|---|---|---|
| Header–Phase 2 | | | | | **121 ± 62** | **35 ± 59** |
| Phase 1–Tank 2 | 0 | **92 ± 52** | **29 ± 30** | | | |
| Phase 1–Tank 3 | 0 | **93 ± 32** | **74 ± 6** | **38 ± 52** | **21 ± 21** | **0** |
| Tank 4 | 0 | **333 ± 58** | **337 ± 70** | **136 ± 136** | | |
| Tank 13 | 0 | **127 ± 76** | **150 ± 41** | **101 ± 59** | | |
| Tank 14 | 0 | **140 ± 77** | **156 ± 41** | **104 ± 36** | **86 ± 44** | **14 ± 23** |
| PS 1 | 0 | **111 ± 46** | **44 ± 42** | **24 ± 24** | **46 ± 43**<br>**404 ± 5** | **0** |
| Jeddah PS | 0 | 77 ± 41 | 38 ± 32 | 0 | | |
| Quiza PS | 0 | 103 ± 40 | 32 ± 18 | 0 | | |
| Mina A PS | 0 | 51 ± 55 | 37 ± 31 | 0 | | |
| Mina B PS | 0 | 78 ± 66 | 21 ± 28 | 49 ± 10 | | |
| Arafa Tank Outlet | 100 ± 141 | **150 ± 265** | **10 ± 12** | 10 ± 15 | | |
| Taif Tank Outlet | 0 | **60 ± 7** | **30 ± 32** | **0** | **0** | |

Increases in nitrate are seen across the system in the trial period compared to the control period, whether or not waters were subjected to chlorination/ammoniation treatment (Table 9). There is no consistent trend of a higher nitrate concentration with the addition of ammonia, but there is some evidence that ammonia addition may be leading to an increase in nitrate in that all tanks have a higher concentration of nitrate on first addition of ammonia in both studies—moving from the control case to condition 1 in this study, just as they did on moving from condition 0 to condition 1 in the 2022 study.

Table 9. Measured nitrate ion concentration (ppb) at locations within the Shoiabah desalination plant site and associated transmission system, 2023. Values **<u>bold and underlined</u>** are locations where water quality should be affected by ammoniation.

| Stage | Control | 1 | 2 | 3 | 4 | 5 |
|---|---|---|---|---|---|---|
| Header–Phase 2 | | | | | **<u>8 ± 8</u>** | **<u>7 ± 12</u>** |
| Phase 1–Tank 2 | **5 ± 3** | **<u>18 ± 15</u>** | **<u>40 ± 34</u>** | | | |
| Phase 1–Tank 3 | **5 ± 3** | **<u>19 ± 8</u>** | **27 ± 12** | **14 ± 11** | **37 ± 14** | **15 ± 15** |
| Tank 4 | **18 ± 10** | **<u>44 ± 17</u>** | **57 ± 26** | **84 ± 7** | | |
| Tank 13 | **19 ± 6** | **<u>23 ± 9</u>** | **39 ± 28** | **68 ± 9** | | |
| Tank 14 | **11 ± 7** | **<u>34 ± 14</u>** | **<u>30 ± 6</u>** | **30 ± 6** | **19 ± 5** | **19 ± 28** |
| PS 1 | **21** | **<u>24 ± 12</u>** | **<u>31 ± 11</u>** | **<u>26 ± 6</u>** | **<u>19 ± 15</u>**<br>**<u>22 ± 13</u>** | **<u>14 ± 22</u>** |
| Jeddah PS | 11 | 44 ± 13 | 34 ± 18 | 38 ± 14 | | |
| Quiza PS | 15 | 20 ± 10 | 39 ± 20 | 74 ± 11 | | |
| Mina A PS | 18 | 23 ± 11 | 29 ± 16 | 41 ± 23 | | |
| Mina B PS | 8 | 16 ± 13 | 15 ± 7 | 9 ± 1 | | |
| Arafa Tank Outlet | 14 ± 8 | **<u>17 ± 11</u>** | **<u>40 ± 11</u>** | **<u>67 ± 24</u>** | | |
| Taif Tank Outlet | 11 ± 3 | **<u>34 ± 9</u>** | **<u>32 ± 14</u>** | **<u>29 ± 15</u>** | **<u>19 ± 24</u>** | |

It should also be noted that as part of the 2023 study, the waters at the Arafa and Taif tank outlets were tested for the presence of organic disinfection products on two dates: on May 25th, when ammoniation was carried out together with the addition of seawater to increase the bromide concentration to approximately 130 ppb, and on June 6th, when enough seawater to increase the bromide concentration to approximately 115 ppb was added. For the ammoniated sample, analysis was conducted for bromodichloromethane, bromoform, and dibromochloromethane, and in all cases concentrations were found to be below the detection threshold of 2 ppb. For the control sample, bromodichloromethane and dibromoacetonitrile were found to be below the detection threshold of 0.10 ppb, while bromoform was identified in the Arafa tank at a concentration of 1.92 ppb and the Taif tank at a concentration of 3.77 ppb, and dibromochloromethane was detected at a concentration of 0.27 ppb in the Arafa tank and 0.33 ppb in the Taif tank. While not as definitive as would be ideal, this indicates that disinfection of water containing 130 ppb bromide did not lead to detectable brominated organic by-product when combined with ammoniation. On June 6th, analysis was also done for monochloracetic acid (detection limit 1.0 ppb), dichloroacetic acid (detection limit 0.5 ppb) and trichloroacetic acid (detection limit 0.5 ppb), with none of these substances detected at either Arafa or Taif tanks, suggesting the treatment did not lead to an increase in haloacetic acids. Comprehensive tests were carried out by a third party for samples collected in August, when temperatures in Makkah province are at a maximum so the rate of generation of bromate and other disinfection by-products should be highest, and water at the Arafa and Taif tanks was found to meet regulatory limits for all disinfection by-products controlled by the Saudi Arabian authorities.

## 5. Conclusions

The combination of chlorination and ammoniation at levels of ammonia as low as 100 ppb has been demonstrated to effectively control the formation of bromate in water produced by seawater desalination on the commercial scale. Preliminary results suggest that the formation of brominate organic disinfection products was also controlled by this treatment. Consistent increases in the proportion of bromine measured as bromide were seen under the same conditions, suggesting that ammonia addition is at least in part controlling bromate formation by reducing the formation of hypobromite intermediate. Trends in nitrate concentration suggest that the addition of ammonia is not contributing significantly to the nitrate load in the product water. Most importantly, bromate control effects of ammonia addition were observed at water storage sites hundreds of kilometres from the seawater desalination plants under summer temperatures, suggesting that the

course of treatment employed will control bromate formation in the transmission lines. This makes it a competitive approach to alternative capital-intensive solutions of second-stage RO to remove bromide at the source or post-treatment adsorption of bromate [49]. Further trials are ongoing within the SWCC network to explore chlorination–ammoniation as a cost-effective method for the control of brominated disinfection by-products.

**Author Contributions:** Conceptualization, A.A.A. (Ali A. Alhamzah); Methodology, A.A.A. (Ali A. Alhamzah), A.S.A. and A.A.A. (Abdulrahman A. Abid); Formal analysis, A.A.A. (Ali A. Alhamzah) and C.M.F.; Investigation, A.S.A. and A.A.A. (Abdulrahman A. Abid); Writing—original draft, C.M.F.; Writing—review & editing, A.A.A. (Ali A. Alhamzah); Project administration, A.A.A. (Ali A. Alhamzah). All authors have read and agreed to the published version of the manuscript.

**Funding:** This research received no external funding and was carried out as part of the authors' duties as employees of the SWCC.

**Data Availability Statement:** The data presented in this study are available on request from the corresponding author. The data are not publicly available due to SWCC policy.

**Conflicts of Interest:** Author Ali A. Alhamzah is named as an inventor on a patent application related to this work. The other authors declare no conflicts of interest.

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
