# Peer review of "Control of Bromate Formation in Desalinated Seawater Production and Transmission with Ammoniation"

_water, doi:10.3390/w15213858_

Round 1

Reviewer 1 Report

Comments and Suggestions for Authors

Manuscript Title : Control of Bromate Formation in Desalinated Seawater Production and Transmission with Ammoniation

After analysing the article, I came to the conclusion that it was difficult to comment since there was an improper flow that made it impossible to study and comprehend the article's content.  The English is really imprecise and hard to comprehend. The conclusion section is missing.  The discussion portion is rather strange. I would want to ask the author to rewrite the article with sincerity and proper planning. The journal WATER has a great reputation. Numerous unrelated references are used; for example, the first paragraph of the introduction has fifteen references. It's unclear what the novelty and theme are.

Comments on the Quality of English Language

English very difficult to understand/incomprehensible

Author Response

We regret any difficulties the reviewer had in comprehending the content of the manuscript. We have taken a number of steps which we hope will improve the comprehensibility of the submission.

  • A paragraph has been inserted early in the introduction to explain the overall flow of the article: This work reports on efforts by the Saline Water Conversion Corporation (SWCC) to minimise formation of bromate and related disinfection byproducts in the water transmitted to consumers by addition of ammonia to produced desalinated water. We will first review and discuss the mechanism of bromate formation in water treated by chlorination and the probable role of ammonia in affecting this mechanism in the light of SWCC’s experience in monitoring water quality. The details of the application of ammonia at a SWCC desalination plant at Shoiabah on the Red Sea and the results of this application will then be presented. The introduction has also been split into two sections, with the title of the second section ‘Mechanisms and Kinetics’ hopefully reflecting better that it is our synthesis of literature data to give a postulated mechanism for ammonia’s role in controlling bromate.
  • We have redrafted many sentences for clarity.
  • We carried out spelling and grammar checks  and accepted all suggestions for improving clarity and conciseness that did not alter the meaning of the text.
  • The previously labelled ‘Discussion’ has been relabelled ‘Conclusion’ to better reflect its content.
  • The ‘Results’ section combines results and their discussion so has been relabelled ‘Results and Discussion’ to better reflect its content.
  • All references included are related to the content of the article. It is not unusual for the first paragraph of a scientific paper to contain a large a number of references. In this particular case, there are a large number of references in the first paragraph to justify the assertion that there has been significant ongoing research interest in a closely-related area: ‘Historically, bromate control has been an issue primarily in surface and ground water treatment with ozonation, and there is a considerable body of research literature addressing control measures for this problem.’

Reviewer 2 Report

Comments and Suggestions for Authors

I found this manuscript worthy of publication. However, I am doubtful for the Schemes. Are the schemes directly copied and pasted from the mentioned references? Because I could find a few "et al." style references in the schemes. If this is so, then it is better to re-design the schemes by taking help from the references, so that the direct overlapping can be avoided.  

Author Response

The schemes were all prepared by the authors. The references to rate coefficients, equilibrium constants, and free energies of reaction on the schemes are cited by author-date rather than reference number as we felt this was more useful to the readers of the article, however we are happy to change the citation style if the editor believes this would be preferable.

Reviewer 3 Report

Comments and Suggestions for Authors

The paper is well represented, however, the introduction part is too lengthy, check for the presentation of Scheme 2

Author Response

We have split the introduction into a brief ‘Introduction’ and a longer section entitled ‘Mechanism and Kinetics’ to reflect that this section is not only a literature review but contains our own calculations and combines reactions reported in different sources into a postulated overall scheme of reactions that may be occurring in our commercial system. Some references to previous work and asides that were not essential have been removed to shorten the text, and we have removed the first part of Scheme 2.

Reviewer 4 Report

Comments and Suggestions for Authors

I found the analysis interesting. Below are some comments that I hope will be helpful in improving the article:

1) A curiosity: is it possible to mention in the introduction or future perspectives pre or post processing techniques that can limit bromate formation? 

2) In the case of low-temperature thermal desalination techniques (e.g., MD; see Coupling of forward osmosis with desalination technologies: System-scale analysis at the water-energy nexus. (2022) Desalination, 543, 116083 and Assessing the validity of solar membrane distillation for disinfection of contaminated water. (2015) Desalination and Water Treatment, 55(10), 2792-2799), what would change regarding the formation of these substances? Does temperature play a key role? Is it possible to control temperature to limit this phenomenon? Perhaps a consideration could be added in the introduction or discussions.

3) How were the errorbars of the various measurements shown in the table estimated? Please add details.

4) I would also mention in the introduction thermal pasteurization techniques for water treatment, e.g.:

Techno-economic analysis of a solar thermal plant for large-scale water pasteurization. (2020) Applied Sciences, 10(14), 4771 

Water disinfection for developing countries and potential for solar thermal pasteurization. Solar Energy, 64(1-3), 87-97.

Author Response

1) A curiosity: is it possible to mention in the introduction or future perspectives pre or post processing techniques that can limit bromate formation?

A sentence has been added to the conclusions giving a brief comparison of the approach to alternatives for controlling bromate and a reference to a very recent review on bromate control in the ozonation case.

2) In the case of low-temperature thermal desalination techniques (e.g., MD; see Coupling of forward osmosis with desalination technologies: System-scale analysis at the water-energy nexus. (2022) Desalination, 543, 116083 and Assessing the validity of solar membrane distillation for disinfection of contaminated water. (2015) Desalination and Water Treatment, 55(10), 2792-2799), what would change regarding the formation of these substances? Does temperature play a key role? Is it possible to control temperature to limit this phenomenon? Perhaps a consideration could be added in the introduction or discussions.

The disinfection by-products form after production of the water, so the temperature of the desalination process per se is not that relevant to their formation; it is the temperature of the storage and transmission that affects the process. Some sentences in the introduction and discussion have been modified or added to clarify this.

3) How were the error bars of the various measurements shown in the table estimated? Please add details.

The measurements reported in the tables in almost all cases are averages of multiple measurements made on successive days under comparable conditions, and the errors are the standard deviation in those sets of measurements. A sentence has been added to the Methods section to clarify this.

4) I would also mention in the introduction thermal pasteurization techniques for water treatment, e.g.:

Techno-economic analysis of a solar thermal plant for large-scale water pasteurization. (2020) Applied Sciences, 10(14), 4771

Water disinfection for developing countries and potential for solar thermal pasteurization. Solar Energy, 64(1-3), 87-97.

Other reviewers have said the introduction was too long, so we have not added discussion of alternative disinfection processes. From the values reported in the 2020 Applied Sciences paper, the cost of thermal pasteurisation would be prohibitively expensive for SWCC.

Round 2

Reviewer 1 Report

Comments and Suggestions for Authors

Accept in present form

Reviewer 4 Report

Comments and Suggestions for Authors

The authors hastily and superficially implemented comments. However, the article can be accepted.